# The Application of Focused Medium-Energy Extracorporeal Shockwave Therapy in Hemophilic A Arthropathy

**DOI:** 10.3390/healthcare10020352

**Published:** 2022-02-11

**Authors:** Wan-Shan Lo, Jiunn-Ming Sheen, Yu-Chieh Chen, Kuan-Ting Wu, Lin-Yi Wang, Yiu-Chung Lau, Chih-Cheng Hsiao, Jih-Yang Ko

**Affiliations:** 1Department of Pediatrics, Kaohsiung Chang Gung Memorial Hospital, Kaohsiung 83301, Taiwan; ilypinky@cgmh.org.tw (W.-S.L.); e5724@cgmh.org.tw (J.-M.S.); gesicht27@cgmh.org.tw (Y.-C.C.); 2Hemophilia Center, Kaohsiung Chang Gung Memorial Hospital, Kaohsiung 83301, Taiwan; 3Chang Gung University College of Medicine, Taoyuan 33302, Taiwan; b9302056@cgmh.org.tw (K.-T.W.); s801121@cgmh.org.tw (L.-Y.W.); benvo@cgmh.org.tw (Y.-C.L.); 4Department of Orthopedic Surgery, Kaohsiung Chang Gung Memorial Hospital, Kaohsiung 83301, Taiwan; 5Center for Shockwave Medicine and Tissue Engineering, Kaohsiung Chang Gung Memorial Hospital, Kaohsiung 83301, Taiwan; 6Department of Physical Medicine and Rehabilitation, Kaohsiung Chang Gung Memorial Hospital, Kaohsiung 83301, Taiwan

**Keywords:** extracorporeal shockwave therapy, hemophilic arthropathy

## Abstract

Hemophilic arthropathy causes the damage of synovium, cartilage, and subchondral bone. The present study evaluated the safety and the effect of extracorporeal shockwave therapy (ESWT), a safe treatment widely used in musculoskeletal conditions in patients with hemophilic arthropathy. Between 1 August 2019 and 31 July 2020, seven hemophilia A patients were enrolled and treated with medium-energy ESWT on the knee joint in the first two months after prophylactic coagulation factor administration. At the beginning of the study and at 1-, 2-, 3-, and 6-month follow-ups, the Hemophilia Joint Health Score (HJHS), visual analog scale score (VAS), and Hemophilia Early Arthropathy Detection with Ultrasound score (HEAD-US) were evaluated for therapeutic effectiveness and safety, while serum bone morphogenetic protein 2 (BMP-2) and von Willebrand factor (vWF) levels were analyzed for assessing chondroprotection and bone healing. Magnetic resonance imaging (MRI) of the knee was performed at the beginning of the study and the 6-month follow-ups. As a result, a non-significant decrease in VAS scores (*p* = 0.151) but not HJHS after treatment was noticed. At the 3-month follow-up, there was a non-significant increase in BMP2 levels (*p* = 0.171) but not vWF. Ultrasonography showed no disease activity score elevation in five patients and no further disease damage in all patients. Repeated MRI examinations in three patients showed no structural progression during the 6-month follow-up. As to adverse events, redness, local heat, and mild swelling were noted in five patients without breakthrough bleeding. We concluded that medium-energy ESWT might be safe for hemophilic arthropathy once prophylactic coagulation factors are administered.

## 1. Introduction

Hemophilia is a congenital coagulation disorder. Its severity can be divided into mild, moderate, and severe depending on the severity of coagulation factor deficiency. Patients with severe hemophilia are severely deficient in coagulation factors, and spontaneous joint or muscle bleeding often occurs, especially in the knee and ankle joints. Repeated episodes of joint bleeding result in joint synovial hyperplasia, cartilage damage, bone destruction, bone spurs, joint swelling, and deformation, and this condition is called hemophilic arthropathy [1,2,3]. Hemophilic arthropathy shares some clinical and biological features with inflammatory joint diseases, such as rheumatoid arthritis (RA)-related synovitis, bone resorption, and degenerative joint diseases, such as osteoarthritis (OA)-related articular cartilage degeneration [1,3].

Pain, stiffness, and limited motion caused by these diseases often cause walking disorders and inconvenience in daily activities. Treatments for hemophilic arthropathy include prophylactic coagulation factors injection, steroid and non-steroidal anti-inflammatory drug (NSAID) administration, rehabilitation, localized steroid injection, isotopic synovectomy, chemical synovectomy, and arthroscopic synovectomy [4,5,6,7]. To date, no preoperative disease-modifying therapy has been available. If the patients’ joint conditions do not respond well to non-surgical management and less-invasive surgical treatments, such as arthroscopic synovectomy, joint arthroplasty has been the surgical treatment of choice. However, surgical treatment for hemophilic patients is a high-risk procedure because these patients have a tendency to bleed during surgery and postoperative wound care. Owing to the aforementioned issues, we aimed to develop a novel, effective, and less-invasive treatment for hemophilic arthropathy.

Extracorporeal shockwave treatment (ESWT) is a non-invasive treatment that is highly safe and has been widely used to treat various musculoskeletal conditions, such as Achilles tendonitis, plantar fasciitis, tennis elbow, and osteoarthritis (OA) [8,9,10,11,12,13,14,15,16]. Based on the known effects and mechanism of ESWT, we expected knee pain and function of the knees in patients with hemophilic arthropathy to be relieved after ESWT. Uncorrected coagulopathy is a contraindication of ESWT. However, despite the improvement in the treatment with coagulation factors, hemophilic arthropathy still progresses early. Owing to the minimally invasive and tissue-protective characteristics of ESWT, we hypothesized that medium-energy ESWT might be beneficial to patients with hemophilic arthropathy and corrected coagulopathy. The primary purpose of this study was to evaluate the safety and efficacy of ESWT for treating hemophilic arthropathy.

## 2. Materials and Methods

We conducted a case-series study of ESWT in patients older than 20 years of age with severe hereditary hemophilia A between 1 August 2019 and 31 July 2020 (IRB number: 201800486B0C603 and 201900290A3C102) at the Hemophilia Center of Kaohsiung Chang Gung Memorial Hospital. These patients met the following inclusion criteria: receiving coagulation factor treatments, having knee pain with a limited response to NSAID treatment, and being diagnosed with hemophilic arthropathy.

Patients were excluded from this study if they were diagnosed with acquired hemophilia or other inflammatory diseases that might affect joints, had received intra-articular injection of hyaluronic acid or steroid treatment previously, had tumors involving the knee or adjacent tissue (benign, malignant, or metastatic), or had acute joint bleeding episodes in the previous week.

All patients received ESWT on the left or right knee and the procedure was performed by a senior specialist. The knee that had been treated with total knee arthroplasty (TKA) was spared from ESWT. The 1- and 2-month follow-up treatments were performed by the same senior specialist. The ESWT site was the musculotendinous junction of the quadriceps tendon. Based on the experiences reported in previous reports, we administered 3000 shockwaves with a medium energy flux density (EFD) of 0.2 mJ/mm^2^ (Duolith SD1^®^, Storz Medical, Tägerwilen, Switzerland) to the knee [14,17,18,19,20], and the treatment duration was 15 min for each session. The patients received prophylactic coagulation factor treatment before each treatment, regardless of their baseline coagulation factor use. The concentration of blood clotting factors was expected to increase by at least 60%.

Assessments with objective and subjective scoring scales and imaging studies were performed to assess the safety and effectiveness of ESWT. The Hemophilia Joint Health Score (HJHS) [21,22] is a measure for evaluating the conditions of bilateral joints in patients with hemophilia and includes the following items: swelling (0 to 3 points), duration (0 or 1 point), muscle atrophy (0 to 2 points), crepitus on motion (0 to 2 points), flexion loss (0 to 3 points), extension loss (0 to 3 points), joint pain (0 to 2 points), and strength (0 to 4 points). A higher score indicates worse clinical conditions. Usually, the score summary includes the scores for the bilateral elbows, knees, and ankles and the global gait score (walking, stair climbing, running, and hopping on one leg). The bilateral joints are evaluated separately. In the present study, only the score for the treated knee was considered. The visual analog scale (VAS) score [23] is a subjective measure of pain that is widely used in clinical practice. The Hemophilia Early Arthropathy Detection with Ultrasound (HEAD-US) evaluation, which is a simplified scanning procedure and scoring method, is used for detecting early hemophilic arthropathy [24]. The HJHS [21,22], VAS score [23], and HEAD-US score [24] were evaluated at the beginning of the study and at 1-, 2-, 3-, and 6-month follow-ups for each patient. In other words, the 2-month HJHS, VAS score, and HEAD-US score were evaluated 1 month after the first ESWT session, while the 3-month follow-up evaluation was performed 1 month after the second ESWT session. The 6-month follow-up evaluation was performed 4 months after completing the second ESWT session. Knee magnetic resonance imaging (MRI) was performed at the beginning of the study and at the sixth month of treatment or later, and the results were interpreted using the International Prophylaxis Study Group (IPSG) MRI scale [25,26].

To evaluate the vascularization, chondroprotection, and bone-healing effects of ESWT, we evaluated the levels of osteogenic growth factors, including the levels of bone morphogenetic protein-2 (BMP-2) and von Willebrand factor (vWF), at the beginning of the study and at 1-, 2-, 3-, and 6-month follow-ups for every patient [18,19,20]. The study process is listed in Figure 1.

Adverse events, including local redness, swelling, and pain, were evaluated using physical examination [17], and an assessment for breakthrough knee joint bleeding was performed using ultrasound examination. The duration of the entire study was 6 months. 

As to statistical analysis, quantitative data are presented as the mean ± standard error (SEM). A repeated-measures analysis of variance test was used for intergroup comparisons. Data analysis was performed by the Statistical Package for Social Sciences (SPSS^®^; IBM Corporation, Armonk, NY, USA) version 22. Descriptive data are presented in the main text, tables, and figures.

## 3. Results

### 3.1. Baseline Characteristics of the Study Population

Eight patients with severe hemophilia A were enrolled. During the first month of the study, one patient was excluded from the study because of acute bleeding of the knee joint before the first ESWT session. The seven patients who completed the study were all men. They received prophylactic recombinant coagulation factor VIII treatment as a baseline treatment for severe hemophilia, with three patients receiving short-acting simoctocog alfa (Nuwiq^®^) treatment (20–40 IU/kg) three days a week, two patients receiving long-acting efraloctocog alfa (Eloctate^®^) treatment (55–65 IU/kg) every five days, and two patients receiving long-acting rurioctocog alfa pegol (Adynovate^®^) treatment (40–50 IU/kg) twice weekly. Four of the seven patients had undergone unilateral total knee replacement surgery before the start of this study.

### 3.2. Safety and Efficacy Evaluations

#### 3.2.1. Hemophilia Joint Health Score (HJHS)

HJHS for the treated knee was calculated at the beginning of the study and at 1-, 2-, 3-, and 6-month follow-ups for every patient. In six out of seven patients (86%), the HJHS for the treated knee did not deteriorate at the latest follow-up visit. The other patient had an elevation of the HJHS by one point after the first ESWT session. However, the score remained unchanged afterwards. Overall, the HJHS did not deteriorate but did not change significantly in the treated knees before and after ESWT (*p* = 0.497, Power = 0.235) (Figure 2).

#### 3.2.2. Visual Analog Scale (VAS) Score

There was a non-significant decrease in VAS score after ESWT. (*p* = 0.151, Power = 0.331) (Figure 3).

#### 3.2.3. Morphogenetic Protein-2 (BMP-2) Levels

There was a non-significant increase in BMP-2 level (*p* = 0.171, Power = 0.455). The levels tended to increase after ESWT until 1 month after the second ESWT session and then seemed to plateau until 4 months after ESWT treatment. (Figure 4).

#### 3.2.4. von Willebrand Factor (vWF) Levels

The vWF levels did not demonstrate significant change after ESWT treatment (*p* = 0.725, Power = 0.151) (Figure 5).

#### 3.2.5. Hemophilia Early Arthropathy Detection with Ultrasound (HEAD-US) Score

The HEAD-US evaluation is composed of two components: evaluation of disease activity (synovitis) scores (0 to 2 points) and disease damage (articular surface or osteochondral damage) scores (cartilage: 0 to 4 points, bone: 0 to 2 points). A higher score indicates more severe disease conditions. Overall, compared to the pre-ESWT disease activity score, by the second month, three patients worsened, and four patients were unchanged. By the sixth month, two patients worsened, two patients improved, and the rest remained unchanged (Figure 6). Regarding the disease damage, all patients’ post-ESWT disease damage status remained unchanged compared to the pre-ESWT status (Table 1).

#### 3.2.6. Knee MRI

The International Prophylaxis Study Group (IPSG) MRI scale was used for the assessment and included soft tissue change and osteochondral change subscores. Three patients (patients No. 1, 2, and 5) completed knee MRI evaluations at the beginning and in the sixth month of the study. Among the three patients, patient No. 1 had a small amount of effusion (or hemarthrosis), a small amount of synovial hypertrophy, and a moderate amount of hemosiderin at the beginning of the study. At the 6-month evaluation, the patient’s effusion (or hemarthrosis) subsided, and the remaining two items remained unchanged (Figure 7). At the beginning of the study and the 6-month evaluation, the other two patients (patients No. 2 and 5) had no effusion (or hemarthrosis), synovial hypertrophy, or hemosiderin. In addition, the osteochondral change subscores in all three patients remained unchanged after two sessions of ESWT. The details are listed in Table 2.

### 3.3. Adverse Events

Among the seven patients who completed the study, no breakthrough knee joint bleeding episodes were found on a series of follow-up ultrasound assessments. Five of the seven patients had local redness, mild swelling, and pain after ESWT, but these conditions subsided within 5–7 days.

## 4. Discussion

The current treatment for patients with severe hemophilia includes regular administration of coagulation factors as prophylaxis rather than on-demand [27,28]. From a global point of view, the high cost of coagulation factors hinders the availability of this treatment. Despite regular administration of coagulation factors, patients with severe hemophilia continue to have spontaneous joint or muscle bleeding. With the increase in the average life span of patients with hemophilia, the aging and degeneration of the joints make hemophilic arthropathy even more complicated. Joint arthroplasty is the treatment of choice once the joint conditions of these patients do not respond well to conservative or less-invasive surgical treatment. However, the underlying coagulopathy increases the bleeding tendency during and after the surgery and affects wound healing, causes periprosthetic joint infection, and affects rehabilitation.

Hemophilic arthropathy shares some clinical and biological features with osteoarthritis (OA), especially OA-related articular cartilage degeneration [1]. ESWT is a non-invasive treatment that is safe and has been widely used in the treatment of various musculoskeletal conditions, including osteoarthritis [8,9,10,11,12,13,14,15,16,17]. Shockwaves are believed to provide many beneficial effects such as pain relief, as well as positive effects on vascularization, protein biosynthesis, cell proliferation, neuro-chondroprotection, and the destruction of calcium deposits in musculoskeletal structures [14,18]. ESWT provides pain relief and enables tissue regeneration by acting directly on nerve fibers and stimulating vascularization [18]. The combination of these treatment effects results in not only the significant alleviation of pain but also the improvement of functional outcomes in injured tissues. Nevertheless, ESWT has contraindications such as uncorrected coagulation abnormalities [17]. Until now, ESWT has had minimal use in patients with hemophilia despite the progress in the treatment with coagulation factors. To our knowledge, there is only one case report describing the application of focused low-energy ESWT in a patient with severe hemophilia A and plantar fasciitis [29]. In the present study, we demonstrated that after appropriate administration of coagulation factors and two sessions of focused medium-energy ESWT, there was no evidence of disease progression on assessments with objective or subjective scoring scales and imaging studies during a 6-month follow-up.

The primary study limitations were the small number of cases and thus the impossibility of conducting it as a double-blinded randomized control trial. This study was only exploratory. To better evaluate the safety and effectiveness of ESWT for hemophilic arthropathy, a large-scale, prospective, double-blinded, randomized control trial enrolling more patients and with a longer follow-up period is needed. In addition, more baseline characteristics of patients with hemophilic arthropathy, including comorbidities and body mass index, should be considered when evaluating the effectiveness of ESWT [27].

Based on published references and clinical experiences of the ESWT performer, side effects of ESWT included pain during ESWT, slight reddening, and superficial hematomata of the skin, which happens even in patients without hemophilia [17]. In the study, ESWT was not performed directly on the knee joints but the musculotendinous junction of the quadriceps tendon. High Energy flux density (EFD) ESWT was avoided as well. According to clinical observation during the study periods, the swelling was minor. It did not cause markable pain or affect the range of motion of the treated knee, different from the manifestations of hemophilic arthropathy. Ultrasound follow-up was performed in addition to clinical observation and reported the absence of breakthrough bleeding. Therefore, we regard the redness, local heat, and mild swelling as side effects as we could expect rather than significant complications.

Energy flux density (EFD) is a parameter based on the flow of shockwave energy through an area with a perpendicular orientation to the direction of propagation, and the unit of measurement is mJ/mm^2^. ESWT has been classified on the basis of EFD as follows: low (<0.08 mJ/mm^2^), medium (<0.28 mJ/mm^2^), and high (<0.60 mJ/mm^2^) treatment intensities. Clinicians typically use an energy of 0.001–0.4 mJ/mm^2^ [18,30,31,32,33]. Treatment with low- and medium-EFD shockwaves triggers the release of nitric oxide (NO), which is beneficial because of its analgesic, angiogenic, and anti-inflammatory effects [8,18]. In previous studies on knee OA treatment, the energy used ranged from 0.25 to 0.4 mJ/mm^2^ [30,31,32,33], and all resulted in favorable therapeutic effects with no severe adverse reactions. Based on this knowledge and owing to the possibility of possible hemorrhagic events related to high-EFD ESWT [34], we administered shockwaves with an EFD of 0.2 mJ/mm^2^.

In our study, a series of follow-up ultrasound examinations did not reveal breakthrough bleeding. Disease activity (synovitis), as assessed by the HEAD-US evaluation, did not deteriorate in five out of seven (71%) patients after two sessions of ESWT. There was no progression in disease damage (articular damage), as assessed by the HEAD-US evaluation, in any of the patients. Three patients underwent MRI evaluations at the beginning of the study and at the 6-month follow-up. There was no worsening of soft-tissue damage, including joint effusion (or hemarthrosis), synovial hypertrophy, or hemosiderin deposition, and there were no osteochondral changes such as surface erosion involving the subchondral cortex or joint margins, subchondral cysts, or cartilage degradation at the latest follow-up visit. The safety of medium-energy ESWT for severe hemophilia A arthropathy was remarkable.

It has been assumed that synovitis and cartilage degradation occur in a continuum as sequential events or dissociative events in hemophilic arthropathy [3,25,35]. In previous studies, Wang et al. reported that ESWT caused neovascularization associated with upregulation of angiogenic and osteogenic growth factors [18,19]. In another study, they demonstrated the regeneration effect of ESWT in patients with hip necrosis with a significant increase in von Willebrand factor (vWF) and VEGF levels [20]. In our study, serum BMP-2 and vWF levels were evaluated to identify the potential chondroprotective and bone-healing effects of ESWT in patients with hemophilic arthropathy [14,15,16]. There was a non-significant increase in BMP2 levels (*p* = 0.171) but not vWF after treatment. The potential of ESWT to have an osteogenic effect or even a bone-healing effect in patients with hemophilic arthropathy might be anticipated but was not able to be demonstrated in this study.

We suggest evaluating hemophilic arthropathic changes by combining HEAD-US and IPSG MRI assessments. These two modalities each have a unique role in the evaluation of hemophilic arthropathy. Ultrasound examination enables a real-time and quick evaluation of joint conditions. However, it was limited by the inability to access the central and load-bearing surfaces because the beam could not penetrate bone boundaries easily. Joint effusion or hemarthrosis was considered a transient phenomenon; thus, it was not included in the HEAD-US scoring system [24]. Moreover, recurrent bleeding episodes cause synovial iron deposition, which is considered an important step in initiating further synovial membrane cell proliferation [1,3,35]. However, this item was not included in the HEAD-US scoring system, and the ability to identify hemosiderin deposition by ultrasound examination is still under debate [24]. Because of the aforementioned limitations, the true and real-time changes in hemophilic arthropathy joints might not be depicted. MRI enabled the identification of early inapparent changes in the joints, but the significance of these early MRI changes has not yet been established. The osteochondral subscores on the IPSG MRI scale reflect gradual joint destruction and were calculated in an additive manner. In contrast, the soft tissue subscores described transient and fluctuating changes. The osteochondral subscores had a ceiling effect and were not useful for discriminating severe changes [25]. A future study involving radiographic examination can overcome this study design-related limitation.

## 5. Conclusions

ESWT with 3000 shockwaves with an EFD of 0.2 mJ/mm^2^ after appropriate coagulation factor administration as a treatment for hemophilic A arthropathy seems to be safe. During the 6-month follow-up, no patients presented structural progression. Further well-controlled, large-scale, and double-blinded randomized prospective studies with an adjustment of ESWT parameters, including frequency, intervals, and energy, may be required to better demonstrate the treatment effect of ESWT in patients with hemophilia A arthropathy.

## Figures and Tables

**Figure 1 healthcare-10-00352-f001:**
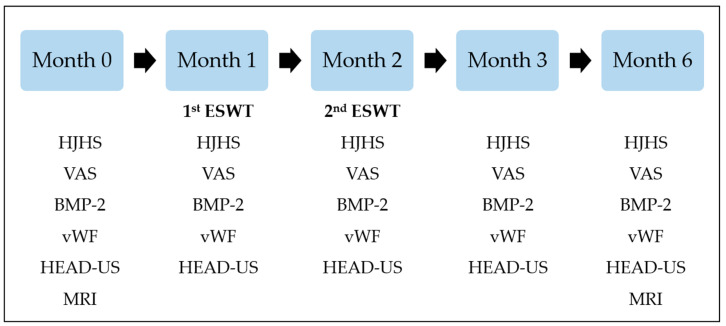
Timetable of the study process. Patients included in the study received ESWT on the left or right knee. ESWT was conducted after the evaluations at 1st and 2nd-month visits and after the prophylactic coagulation factor administration. The Hemophilia Joint Health Score (HJHS) of the treated knee, the visual analog scale (VAS) score, the Hemophilia Early Arthropathy Detection with Ultrasound (HEAD-US) evaluation of the treated knee, the levels of bone morphogenetic protein-2 (BMP-2), and von Willebrand factor (vWF) were evaluated at the beginning of the study and 1-, 2-, 3-, and 6-month follow-ups for each patient. The knee magnetic resonance imaging (MRI) was performed at the beginning of the study and the 6-month of treatment or later, and the results were interpreted using the International Prophylaxis Study Group (IPSG) MRI scale.

**Figure 2 healthcare-10-00352-f002:**
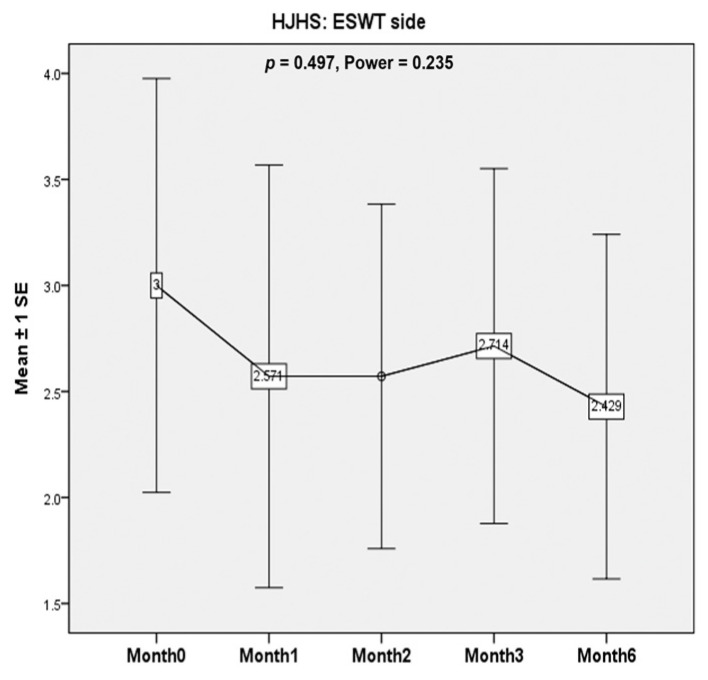
The Hemophilia Joint Health Score (HJHS). The Hemophilia Joint Health Score (HJHS) is a measure for evaluating the conditions of bilateral joints in patients with hemophilia and includes the following items: swelling (0 to 3 points), duration (0 or 1 point), muscle atrophy (0 to 2 points), crepitus on motion (0 to 2 points), flexion loss (0 to 3 points), extension loss (0 to 3 points), joint pain (0 to 2 points), and strength (0 to 4 points). A higher score indicates worse clinical conditions. Usually, the score summary includes the scores for the bilateral elbows, knees, and ankles, and the global gait score (walking, stair climbing, running, and hopping on one leg). The bilateral joints are evaluated separately. In the present study, only the score for the treated knee was considered. Overall, the HJHS did not deteriorate but did not change significantly in the treated knees before and after ESWT (*p* = 0.497, Power = 0.235).

**Figure 3 healthcare-10-00352-f003:**
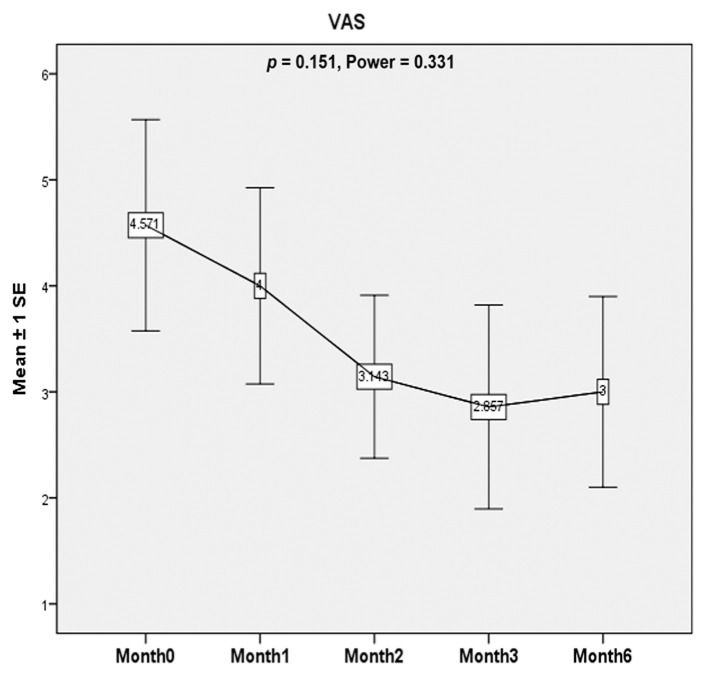
Visual analog scale (VAS). Patients make a mark on a 10 points scale that corresponds to the intensity of pain. Higher points indicate an increasing degree of pain. There was a non-significant decrease in VAS score after ESWT. (*p* = 0.151, Power = 0.331).

**Figure 4 healthcare-10-00352-f004:**
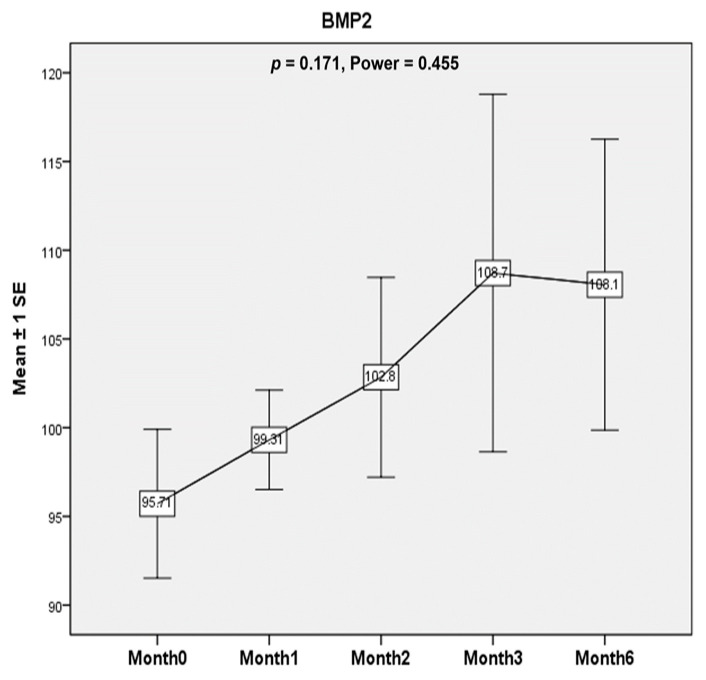
Bone morphogenetic protein-2 (BMP-2). Serum BMP-2 level was evaluated to identify the potential chondroprotective and bone-healing effects of ESWT in patients with hemophilic arthropathy in this study. The BMP-2 levels tended to increase after ESWT until 1 month after the second ESWT session and then seemed to plateau until 4 months after ESWT treatment. However, the results were not statistically significant (*p* = 0.171, Power = 0.455).

**Figure 5 healthcare-10-00352-f005:**
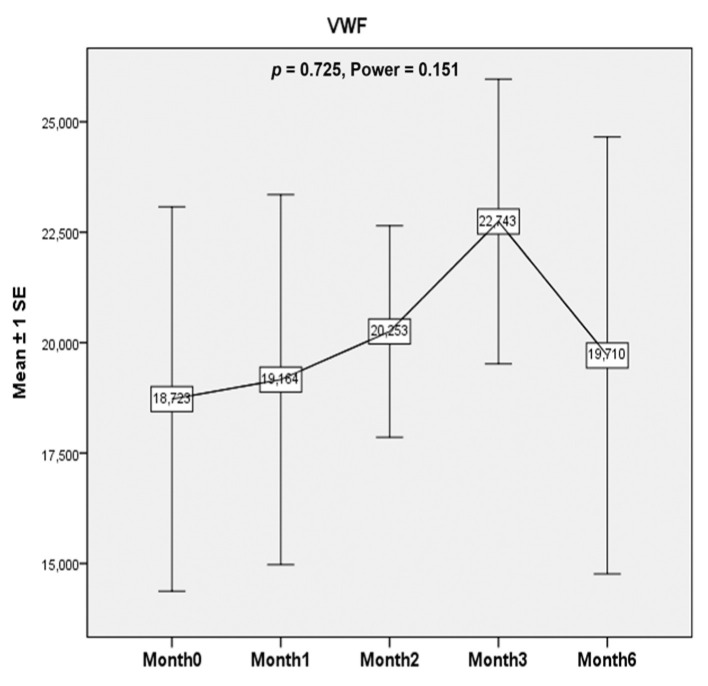
von Willebrand factor (vWF). Serum vWF level was evaluated to identify the potential bone-healing effects of ESWT in patients with hemophilic arthropathy in this study. The vWF levels did not demonstrate significant change after ESWT treatment (*p* = 0.725, Power = 0.151).

**Figure 6 healthcare-10-00352-f006:**
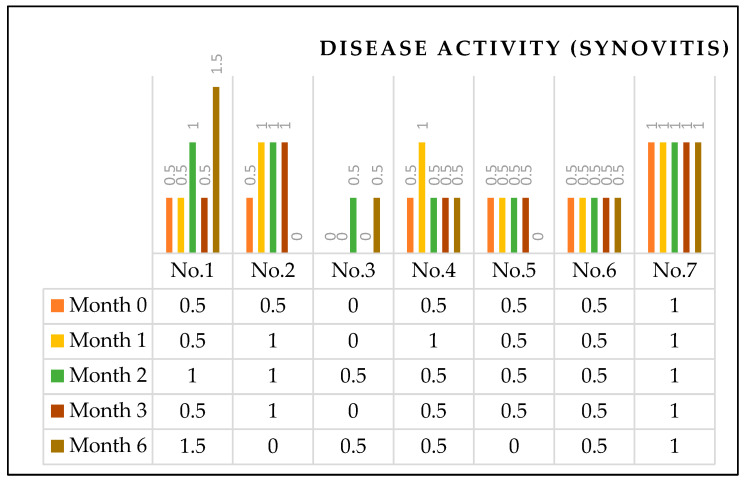
Disease activity (synovitis) scores of HEAD-US. The HEAD-US evaluation is composed of two components: evaluation of disease activity (synovitis) scores (0 to 2 points) and disease damage (articular surface or osteochondral damage) scores (cartilage: 0 to 4 points, bone: 0 to 2 points). A higher score indicates more severe disease conditions. Overall, compared to the pre-ESWT disease activity score, by the second month, three patients worsened, and four patients were unchanged. By the sixth month, two patients worsened, two patients improved, and the rest remained unchanged.

**Figure 7 healthcare-10-00352-f007:**
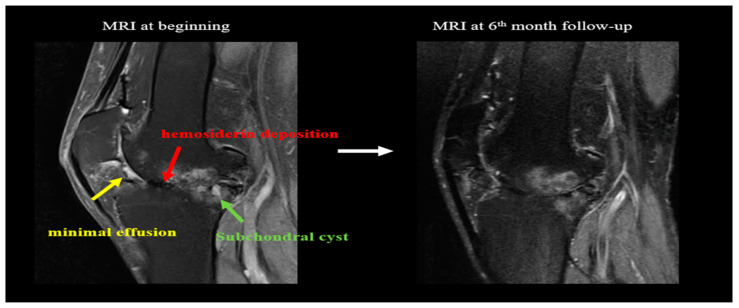
MRI before ESWT and after two sessions of ESWT. The Figure demonstrates No. 1 patient’s PD FS (protein density fat suppression) MRI images at the beginning of the study and at the 6-month follow up. A small amount of effusion (or hemarthrosis), synovial hypertrophy, and a moderate amount of hemosiderin were noticed at the beginning. At the 6-month evaluation, the patient’s effusion (or hemarthrosis) subsided. The synovial hypertrophy and hemosiderin deposition remained similar without progression. On the other hand, osteochondral changes have existed since the beginning of the study, including markable surface erosion involving the subchondral cortex and joint margins, subchondral cysts, and cartilage degradation. These osteochondral changes remained similar at the 6-month follow-up.

**Table 1 healthcare-10-00352-t001:** Disease damage (articular surface) scores of HEAD-US *.

Patient\Time	Month 0	Month 1	Month 2	Month 3	Month 6
No. 1	6	6	6	6	6
No. 2	2	1	1	0	1
No. 3	1	0	0	0	0
No. 4	1	1	0	1	1
No. 5	0	0	0	0	0
No. 6	0	0	0	0	0
No. 7	2	1	0	1	1

* The HEAD-US evaluation is composed of two components: evaluation of disease activity (synovitis) scores (0 to 2 points) and disease damage (articular surface or osteochondral damage) scores (cartilage: 0 to 4 points, bone: 0 to 2 points). A higher score indicates more severe disease conditions. Regarding the disease damage, all patients’ post-ESWT disease damage status remained unchanged compared to the pre-ESWT status.

**Table 2 healthcare-10-00352-t002:** International Prophylaxis Study Group (IPSG) MRI scale assessments of hemophilic arthropathy results.

Categories			Points	No. 1		No. 2		No. 5	
				M0	M6	M0	M6	M0	M6
Soft tissue changes	Effusion/hemarthrosis	Small	1	V					
		Moderate	2						
		Large	3						
	Synovial hypertrophy	Small	1	V	V				
		Moderate	2						
		Large	3						
	Hemosiderin	Small	1						
		Moderate	2	V	V				
		Large	3						
Soft tissue changes subscore (Maximum 9 points)				4	3	0	0	0	0
Osteochondral change	Surface erosion involving subchondral cortex or joint margins								
	Any surface erosion		1	V	V				
	Half or more of the articular surface eroded in at least one bone		1	V	V				
	Subchondral cysts								
	At least one subchondral cyst		1	V	V				
	Subchondral cysts in at least two bones, or cystic changes involving a third or more of the articular surface on at least one bone		1	V	V				
	Cartilage degradation								
	Any loss of joint cartilage height		1	V	V	V	V		
	Loss of half or more of the total volume of joint cartilage in at least one bone		1	V	V	V	V		
	Full thickness loss of joint cartilage in at least some area in at least one bone		1	V	V				
	Full thickness loss of joint cartilage including at least one half of the joint surface in at least one bone		1	V	V				
Osteochondral change subscore (Maximum 8 points)				8	8	2	2	0	0
Total score (Maximum value 17 points)				12	11	2	2	0	0

## Data Availability

The datasets used and/or analyzed during the present study are available from the corresponding author upon reasonable request.

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
