# Peer review of "The Application of Focused Medium-Energy Extracorporeal Shockwave Therapy in Hemophilic A Arthropathy"

_healthcare, 2022, doi:10.3390/healthcare10020352_

Round 1

Reviewer 1 Report

The application of focused medium-energy extracorporeal shockwave therapy in hemophilic A arthropathy

In this Manuscript, Lo et al. report the data from an open, non-randomized clinical trial evaluating the use of extracorporeal shockwave therapy in hemophilic A chronic arthropathy resistant to NSAID.

The novelty of the study is important in a rare disease making it a valuable contribution to the field. However, several aspects should be improved to warrant publication.

1/ Introduction :

The rationale of using ESWT in hemophilia A chronic arthropathy should be better described in the introduction because its use is typically contra-indicated in patients with blood disorders. This specific contra-indication (with risk of hemarthrosis and disease worsening) should be discussed early on in the introduction and is a major objective of the study.

  • prophylaxis injection line 55 is unclear, do the author mean coagulation factor ? Antifibrinolytic therapy ? Please clarify
  • Citing a review on the management of HA arthropathy would be useful here.

2/ Methods :

  • Please provide a small figure showing the design (timings of ESWT therapy and evaluations (ie, MRI, US, clinical)).

3/ Results of the study :

All study results only show non-statistical difference, which is expected (and is okay) considering the number of patients included. However :

  • The authors should not oversell too much their results, especially in the abstract. Moreover, most scientist agree to say that the term trend shoud be banned. For example : there is a non-significant BMP2 increase (p = 0.17), but not for vWF (p =0.7). The abstract could say something like : “there was a non-significant increase in BMP2 (p = 0.171) but not vWF after treatment.”
  • Same in the abstract for VAS and HJHS : state “non-significant decrease” for VAS. For HJHS, the p = 0.497 which is hardly a decrease but rather a stability.
  • It is ABSOLUTELY necessary that the author report in the abstract the adverse events (ie, pain, swelling +/- hemarthrosis) following ESWT, since this is the main objective of their study (evaluate safety). Indeed the authors report 5 cases of swelling following ESWT, did the authors investigate this swelling ?
  • Considering the HEAD-US score, there is 2 components : the damage which can theorically only WORSEN since we don’t know how to heal cartillage; and the disease activity, which can be improved by treatment. I suggest the author better describe the latter. Maybe showing the synovitis score in a figure would be more informative. By month 2, 3 patients worsened, and 4 patients were unchanged. By month 6, 2 patient worsened, 2 patients improved and the rest remained unchained. Please clarify this part of the results section.
  • Line 35-36 : “Repeated MRI examinations in three patients 35 showed no progression of the previous pathologies”. I believe the authors mean no “structural progression”.
  • Please number the patients 1-7 thorough the results section, so we know which patients underwent MRI : are these the patients who responded better by US evaluation ?

3/ Discussion / Conclusions :

The authors should underline these results are exploratory since it is an open/uncontrolled/non-randomized study.

  • Particularly, in the abstract, the authors cannot state that this intervention “seemed to protect hemophilia A arthropathy from deterioration”. All they can state is that during the 6 month follow-up no patient presented structural progression since there is no control group.
  • In the main discussion the authors should state the limitation stated earlier (open/uncontrolled/non-randomized). To demonstrate efficacy controlled randomized double blind study are needed and this study is only exploratory.

Minor points:

  • The authors should provide molecule names of the drugs given to the patients.

Author Response

Thank you very much for your kind comments and suggestions. We want to show our gratitude for your thorough consideration for the study and article and gave us the opportunity for revision. Please see the attachment for our point-by-point modifications and response to each of your kind suggestions and comments.

Sincerely yours

Reviewer 2 Report

  1. Did anyone develop bleeding after 6 months?
  2. Why the study included the patients that have undergone knee surgery and what type of improvement were noticed in these patients after extracorporeal therapy?
  3. In the MRI , it looks like there was no improvement of the finding, so will this therapy is a symptomatic therapy only?

Author Response

Thank you very much for your kind comments and suggestions. We want to show our gratitude for the opportunity to revise this article. Please see the attachment for our point-by-point response to each of your kind comments.

Sincerely yours

Round 2

Reviewer 1 Report

The authors have made an excellent job at revising their manuscript.

Please just add a legend (with all abbreviation) to newly-added figure 1 describing the timetable of the study so the figure can be read without going to the text

Author Response

We are grateful to learn all your comments and suggestion to perfect the article. Thank you very much for the suggestions of adding legend (with all abbreviation) to newly-added figure 1 describing the timetable of the study. It was indeed our mistake. The revision has been made as suggested. Please see lines 139-151.